# Wasting and Associated Factors among Children under 5 Years in Five South Asian Countries (2014–2018): Analysis of Demographic Health Surveys

**DOI:** 10.3390/ijerph18094578

**Published:** 2021-04-26

**Authors:** Nidhi Wali, Kingsley E. Agho, Andre M. N. Renzaho

**Affiliations:** 1School of Social Sciences, Western Sydney University, Locked Bag 1797, Penrith, NSW 2751, Australia; 2School of Health Sciences, Campbelltown Campus, Western Sydney University, Locked Bag 1797, Penrith, NSW 2571, Australia; K.Agho@westernsydney.edu.au; 3African Vision Research Institute, Westville Campus, University of KwaZulu-Natal, Durban 3629, South Africa; 4School of Medicine, Campbelltown Campus, Western Sydney University, Locked Bag 1797, Penrith, NSW 2571, Australia; Andre.Renzaho@westernsydney.edu.au; 5Translational Health Research Institute, Western Sydney University, Locked Bag 1797, Penrith, NSW 2571, Australia; 6Maternal, Child and Adolescent Health Program, Burnet Institute, Melbourne, VIC 3004, Australia

**Keywords:** child undernutrition, factors, infants, wasting, South Asia

## Abstract

Child wasting continues to be a major public health concern in South Asia, having a prevalence above the emergency threshold. This paper aimed to identify factors associated with wasting among children aged 0–23 months, 24–59 months, and 0–59 months in South Asia. A weighted sample of 564,518 children aged 0–59 months from the most recent demographic and health surveys (2014–2018) of five countries in South Asia was combined. Multiple logistic regression analyses that adjusted for clustering and sampling weights were used to examine associated factors. Wasting prevalence was higher for children aged 0–23 months (25%) as compared to 24–59 months (18%), with variations in prevalence across the South Asian countries. The most common factor associated with child wasting was maternal BMI [adjusted odds ratio (AOR) for 0–23 months = 2.02; 95% CI: (1.52, 2.68); AOR for 24–59 months = 2.54; 95% CI: (1.83, 3.54); AOR for 0–59 months = 2.18; 95% CI: (1.72, 2.77)]. Other factors included maternal height and age, household wealth index, birth interval and order, children born at home, and access to antenatal visits. Study findings suggest need for nutrition specific and sensitive interventions focused on women, as well as adolescents and children under 2 years of age.

## 1. Introduction

Wasting, weight-for-height (WHZ) *<* −2 of WHO’s child growth standards, in children poses a serious threat to child survival and development, with the heightened risk of mortality [1]. Child wasting can result in adverse and often irreversible consequences, including poor cognition and learning performance, reduced lean body mass, short adult stature, lower productivity, and reduced earning [2,3]. While wasting levels have declined slowly over the past 40 years, globally, there continues to be around 58.3 million (in 2017) wasted children under 5 years, the majority of whom reside in South Asian countries [4]. The prevalence of child wasting in South Asia is above the 15% threshold [5]. Furthermore, child wasting in South Asia has several unique characteristics when compared to other regions. These include high prevalence and incidence of wasting at birth and in early life, prolonged periods of wasting experienced by children in the first two years of life, and higher prevalence of concurrent stunting and wasting [6]. All these characteristics establish child wasting as a ‘critical public health problem’ in the South Asia region, making it the global epicentre of child wasting [5].

Unravelling the wasting conundrum in South Asia remains a challenge. However, the UNICEF conceptual framework provides a framework to better conceptualise factors associated with wasting in children under 5 years. The framework identifies an interplay of multiple factors at immediate, underlying, and basic levels. The immediate factors include low energy and nutrients intake, nutrient losses due to infection, or a combination of both low energy or protein intake and high nutrient loss by the mother during pregnancy or by the child after birth [7]. The significant underlying factors include maternal characteristics of BMI, height, and education, as well as socio-economic status [8,9,10,11], and access to services, including water sanitation and hygiene services [10]. For instance, a study in Nepal found child’s age and early initiation of breast feeding were associated with child nutrition status [8] while another study in India observed age, gender, birth order, place of residence (rural), household income, and mother’s age were significantly associated with child wasting [11]. A study in Bangladesh using data from the demographic health survey identified low birth weight in children as a key determinant for child wasting and mother’s education, socio-economic status, and birth interval as other factors [9]. Another population based cross-sectional study conducted in six South Asian countries identified that younger children (0 to 5 months) and those whose mothers had a low body mass index (<18.5 kg/m^2^) were common risk factors for child wasting across all countries [12].

South Asia presents a paradox, also commonly known as the ‘South Asia enigma’, a term used to describe persistent high levels of child undernutrition despite increased economic growth [13]. The region’s economic growth [14,15,16] has not effectively translated into reduction in child undernutrition, including wasting. While it is the fastest growing developing region in the world with significant economic progress, marked poverty reduction, improved indicators of health, literacy, and agriculture outputs [14,15,16], it also hosts the largest burden of global child undernutrition. Child undernutrition can be a major economic burden for the South Asia region. Moving forward, understanding factors associated with child wasting and severe wasting and addressing them becomes not only vital but also cost effective. There is an estimated cost of US$200 attached to treating each severely wasted child [17]. The 2013 Lancet series on undernutrition recognized treatment of severe acute malnutrition as the most cost effective of the various direct nutrition interventions [18].

This study aims to understand factors most significantly associated with child wasting and severe wasting for children aged <5 years; more specifically, it aims to understand the association of the region’s economic progress with child wasting. The reasons for doing so are threefold: presently, there is limited evidence to understand specific determinants of child wasting when compared to other forms of child undernutrition, such as child stunting, specifically to understand and explore the association between household wealth and child wasting; Second, most of the studies in the region have predominantly focused on children aged <5 years [8,9,10,11]. However, growing evidence indicates that child wasting majorly occurs during the 1000-day period that spans from conception until 2 years of child’s age [19]. The prevalence of child wasting is more than two times higher among children 0–23 months than among children 24–59 months (9·2 vs. 3·8%), while the prevalence of severe wasting is almost four times higher (3·8 vs. 1·0%) [2]. Finally, most of these studies have been country specific, except for a population based cross-sectional study conducted in five SA countries that looked at children aged <5 years but did not examine child wasting among different age groups, specifically children from 0–23 months, despite the high prevalence in this age group [12].

Our study builds upon these limitations by pooling data from five SA countries using the most recent DHS datasets (2014–2018). By pooling DHS data that include all children aged <5 years, our analysis permits enhanced statistical power to address inconsistencies in the current evidence to identify sources of diversity across various DHS datasets in the region and to compare outcomes models across settings. Findings from the study will contribute to the growing body of literature to support the prioritization of child wasting, in addition to ongoing efforts to reduce child undernutrition in the region. Evidence from the pooled analysis will help identify common intervention for wasting for children with common characteristics such as culture and dietary patterns. Findings from the study will enable understanding of factors concerning all children aged <5 years and generate evidence to inform future policy and region-specific targeted interventions.

## 2. Materials and Methods

This study utilized datasets from the most recent 2014–2018 demographic health survey (DHS) conducted in countries within the South Asia region, including Bangladesh, India, Nepal, Maldives, and Pakistan. Data for other South Asian countries were not available through DHS due to the following reasons: Afghanistan does not collect anthropometric data for children under 5 years of age, data for Bhutan is unavailable on DHS, and finally, data for Sri Lanka have restricted access and are not publicly available for research purposes. The DHS is a nationally representative survey that collects data on mortality, fertility, family planning, and maternal and child health [20]. The DHS programme uses standardized methods in their surveys to ensure uniformity of results from different countries. These surveys were comparable, given the standardized nature of the data collection methods and instruments [20]. Data were obtained from a password-enabled Measure DHS website [21].

Information was collected from eligible women, that is, all women aged 15–49 years who were either permanent residents in the households or visitors present in the households on the night before the survey. Child health information was collected from the mother based on the youngest child aged less than five years, with response rates that ranged from 96% to 99%. Detailed information on the sampling design and questionnaire used is provided in the respective country-specific Measure DHS reports [21]. Our analyses were restricted to 564,518 children aged 0–59 months for five South Asian countries.

### 2.1. Study Variables

The outcome variable was wasting (low weight-for-height). Wasting measures body mass in relation to height and describes current nutritional status. Based on the 2007 WHO growth reference, children with weight-for-height Z-scores below minus two standard deviations (-2 SD) below the mean of WHO child growth standards are considered wasted or acutely malnourished while children with Z-scores below minus three standard deviations (-3 SD) below the mean of WHO child growth standards are considered severely wasted [22,23].

### 2.2. Potential Confounding Factors

The choice of confounding factors used in this study was informed by the UNICEF framework [7]. The confounding factors were organised into three groups: (i) *Immediate factors:* dietary diversity score and child’s disease occurrence (episodes of diarrhoea and fever in the last two weeks); feeding practices, such as currently breastfeeding and duration of breastfeeding; vitamin A supplementation; vaccination coverage; and child’s age and sex. (ii) *Underlying factors:* including *mother’s characteristics,* such as age; age at birth; height; BMI; marital status; birth order and interval; maternal and paternal education; women’s power over household earnings, household decision-making, and health care autonomy. *Household factors:* pooled household wealth index, access to source of water, and type of toilet, which was categorised into improved and unimproved sources. *Access and Utilisation of services:* healthcare utilisation factors, such as place and mode of delivery; combined birth rank (the position of the youngest under-five child in the family) and birth interval (the interval between births; that is, whether there were no previous births, births less 24 months prior, or births more than or equal to 24 months prior); delivery assistance; antenatal clinic visits (ANC); and access to media services, such as listening to the radio, watching television, and reading newspapers or magazines. (iii) *Basic factors:* such as country and place of residence (urban or rural). In order to reduce collinearity, we combined place of birth and mode of delivery and birth order and birth interval. The combined mode of delivery and place of birth was divided into three categories as delivered at home, delivered at a health facility with non-caesarean section, and delivered at a health facility with a caesarean section while the combined birth order and the birth interval was classified as birth rank and birth interval because of the collinearity between birth order and birth interval, which is consistent with previous studies [24,25]. Maternal height was divided in the five following categories: <145 cm, 145–149.9 cm, 150–154.9 cm, 155–159.9 cm, and ≥160 cm, with <145 cm defined as short maternal height [26].

The household wealth index factor score (hv271) was constructed by DHS for each country. For each country, the hv271 variable used that principal component’s statistical procedure, which was used to determine the weights for the wealth index based on information collected about 22 household assets and facilities and produce the standardised scores (z-scores) and factor coefficient scores (factor loadings) of wealth indicators. The household wealth index factor score for the pooled dataset was constructed using the ‘hv271’ variable. The household wealth index factor score (hv271) was separated into categories, as the bottom 20% of households were arbitrarily referred to as the poorest households and the top 20% as the richest households, and was divided into poorest, poor, middle, rich, and richest.

Dietary diversity (DD) was calculated by summarizing the 7 food groups consumed during the last 24 h. These foods are grains, roots, and tubers; legumes and nuts; milk/dairy products; flesh foods (meat, fish, poultry and liver/organ meats); vitamin-A rich fruits and vegetables; other fruits and vegetables; and eggs. The children were categorised into two groups, namely, the child had ≥4 food groups and the child had <4 food groups [27].

### 2.3. Statistical Analysis 

To examine factors associated with wasting among children aged 0–23 months, 24–59 months, and children 0–59 months, the dependent variables were expressed as a binary outcome, i.e., category 1 for wasting (<−2SD) and otherwise category 0. For the combined five South Asian countries, a population-level weight, unique country-specific clustering, and strata were created to avoid the effect of countries with a large population (such as India with over 1.4 billion people in 2017 [28], offsetting countries with a small population (such as the Maldives with about 437,535 people in 2017) [29]. Population-level weights were used for survey (svy) tabulation that adjusted for a unique country-specific stratum, and clustering was used to determine frequency tabulations to describe the characteristics of the study population and descriptive analysis for estimating 95% confidence intervals (CI) around prevalence estimates for wasting among children aged 0–23 months, 24–59 months, and 0–59 months in each country.

Univariate logistic regression analyses were used and presented as unadjusted OR (95% CI) for each confounding variable, while multivariate logistic regression was used to identify independent factors associated with wasting among children aged 0–23 months, 24–59 months, and 0–59 months after adjusting for clustering and sampling weights.

In the multivariate logistic regression analyses, three-stage modelling was employed. In the first stage, the immediate factors were entered into the first stage model (Model 1), and a manually executed elimination method was used to determine factors associated with wasting at *p* < 0.05. The significant factors in the first stage model were then added to underlying factors, which was the second stage model (Model 2); this was then followed by manually executed elimination procedure. The significant factors for both Models 1 and 2 were added to basic factors in the third model (Model 3). We used a similar approach for basic factors in (Model 3). Factors associated with wasting among children aged 0–23 months, 24–59 months, and 0–59 months were presented as adjusted OR (95% CI) for the variables retained in the final modelling step. These results are presented in Tables 2–4 in the results section. These analyses were performed using Stata ‘svy’ commands that allow for adjustments of country-specific stratum and population-level weights. STATA V.14.1 (STATA Corporation, College Station, TX, USA, 2015) was employed for all the study analyses.

## 3. Results

The demographic profile of the sample is presented in Table 1.

### 3.1. Prevalence of Wasting in Children Aged 0–23 Months, 24–59 Months and 0–59 Months 

Figure 1 shows prevalence and 95% CI of wasting in children aged 0–23 months in South Asia by country. As illustrated in Figure 1, the prevalence of wasted children aged 0–23 months was highest in India (26%), followed by Bangladesh (17%), Nepal (15%), Pakistan (10%), and lowest in Maldives (7%). The overall pooled prevalence of wasted children aged 0–23 months in five South Asian countries was 25%.

Figure 2 shows prevalence and 95% CI of wasting in children aged 24–59 months in South Asia by country. As illustrated in Figure 2, the pooled prevalence of wasting among children aged 24–59 months in five South Asian countries was 18%, with India reporting the highest prevalence of wasting at 18%, followed by Maldives at 11%, Bangladesh at 13%, Nepal at 6%, and lowest in Pakistan at 5%.

Figure 3 shows prevalence and 95% CI of wasting in children aged 0–59 months in South Asia by country. Pooled prevalence of wasting in children aged 0–59 months in South Asia (Figure 3) was 20%. India had the highest prevalence (21%) of wasting amongst children aged 0–59 months. The prevalence of Bangladesh was at 14%, Nepal at 10%, Maldives at 9%, and lowest in Pakistan at 7%.

### 3.2. Factors Associated with Child Wasting for Children Aged 0–23 Months

Table 2 describes the factors associated with wasting in children aged 0–23 months in five South Asian countries. Wasting among children aged 0–23 months was associated with being born in India, having a mother who is short, 145–149 cm, and underweight (BMI ≤ 18.5 kg/m^2^), children who were born on 4th birth rank with more than two years interval, and being delivered through vaginal birth at a health facility and those born at home. Children from poorer households and from middle class household were less likely to be wasted than those from the poorest households.

### 3.3. Factors Associated with Child Wasting for Children Aged 24–59 Months

Table 3 describes the factors associated with wasting in children aged 24–59 months in five South Asian countries. Wasting among children aged 24–59 months was associated with being born in India. Mothers with normal BMI (18.5–25 kg/m^2^) and underweight mothers (BMI ≤ 18.5 kg/m^2^) were more likely to have wasted children than those who were obese. Mothers who made no ANC visits were less likely to be wasted compared with those mothers who made eight or more ANC visits. Children aged 24–59 months born to mothers aged 25–34 years and 35–49 years reported an increased odd of wasting than children born to mothers aged 15–24 years. Children born in richer households were less likely to be wasted.

### 3.4. Factors Associated with Child Wasting for Children Aged 0–59 Months

Table 4 describes the factors associated with wasting in children aged 0–59 months in five South Asian countries. Wasting among children aged 0–59 months was associated with being born in India, maternal height (155–159 cm), having a mother with normal BMI (18.5–25 kg/m^2^) and underweight mother (BMI ≤ 18.5 kg/m^2^). Children delivered at a health facility through vaginal birth were more likely to be wasted than those children delivered through caesarean. Mothers who made no ANC visits during pregnancy were less likely to be wasted than those mothers who made eight or more ANC visits during pregnancy. Children from poorer households and from richer households were less likely to be wasted than those from poorest households. Children who had adequate dietary diversity were less likely to be wasted. Children aged 24–59 months had lower odds of being wasted than younger children aged 0–23 months.

## 4. Discussion

Our findings revealed the overall pooled prevalence of wasted children in five South Asian countries (2014–2018) was 25% for children aged 0–23 months, 18% for children aged 24–59, and 20% children aged 0–59 months. Wasting prevalence in South Asia constitutes a ‘critical public health’ emergency with wasting levels above 15% in the region [5]. Our research found that children <5 years of age were most likely to be wasted if they lived in India, while prevalence was high for those living in India, Bangladesh, and Maldives. Addressing child wasting in South Asia can significantly reduce the global burden of wasting, as the region hosts more than half of all the world’s wasted children, as supported by other research in the region [12]. The wasting prevalence rates in South Asia demand for an increased and simultaneous focus on implementing nutrition-specific and nutrition sensitive interventions [18], which will enable addressing the immediate and underlying factors of child undernutrition.

Our study examined wasting and associated factors among different age groups of all children under 5 years: 0–23 months, 24–59 months, and 0–59 months, because intervention to improve nutrition and health knowledge and practices including psychosocial stimulation interventions for wasted children should be targeted at 0–23 months and 0–59 months, respectively [30]. Study findings suggest that wasting was higher among children aged 0–23 months than those aged 24–59 months. It points out that infants are at higher risk to wasting during the first two years, which significantly reduces with age. This finding is aligned with an emphasis on the 1000 days of conception to two years of a child’s life as the critical window of opportunity where substantial impacts can be achieved on child physical growth and brain development [31,32]. Proper development in the first two years of a child’s life can determine the nutrition and health status for their entire life.

In our study, wasting prevalence for children aged 0–23 months in all countries was above the emergency threshold, except Maldives. The wasting prevalence in Maldives, although lowest at 7% amongst other South Asian countries, was still high for a LMIC [5]; while for children aged 24–59 months, Maldives had a prevalence of 11%, which is above the medium threshold [5]. Research in Maldives suggests the changing dietary patterns, with limited intake of fresh fruits and vegetables and increased dependency on imported packaged foods, has contributed to an increased burden of malnutrition amongst children <5 years [33]. However, further research is required to understand these variations across South Asian countries.

Children born in poorest and poorer households were at higher odds of being wasted as compared to wealthier households across all children <5 years. A research in 35 LMICs found high prevalence of wasting in children in poorest households [26]. Research in South East Asia suggests that children from the poorest wealth quintiles had 25% increased risk of being wasted when compared to the richest wealth quintile [34]. Households with low income tend to spend less on proper nutrition and are more susceptible to growth failure due to poor access to sufficient food of adequate quality and poor living conditions. Our research did not find a pattern of wasting in relation to improved water and sanitation and maternal education, as identified by previous research [12,26,34]. In our adjusted models, these factors were not consistently associated with wasting across South Asia. This could be due to improved literacy rates [14,15,16], along with improved access to water and sanitation facilities [35] in the region.

Mother’s height was associated with child wasting; maternal heights of 145–149 cm for 0–23 months and 155–159 cm for 0–59 months had higher odds of having wasted children. Previous research shows association of short maternal stature and child wasting amongst children aged 0–59 months. A study in Bangladesh using the DHS data showed that children <5 years of short statured mothers (<145 cm) were at 1.28 times the risk of wasting and 1.43 times the risk of severe wasting than tall mothers [36]. Another study in India showed similar associations [37]. Research in 54 low-income countries showed that every one-centimetre increase of maternal height significantly reduced the risk of wasting among children <5 years [38]. Our study further points out that this association is statistically more significant for children aged 0–23 months when compared to older children (24–59 months). Maternal height is an important indicator of a mother’s cumulative net nutrition and biological deprivation over periods of rapid growth; it allows for assessment of intergenerational linkages in the child’s health before or immediately after birth with lasting influence over a few years. Underweight mothers (BMI ≤ 18.5 kg/m^2^) and older mothers (25 years or older) had higher odds of having wasted children. Maternal BMI, including underweight mothers (BMI ≤ 18.5 kg/m^2^) and those within the normal range (18.5–25), was associated with higher odds of wasting across all children aged <5 years. This finding is consistent with previous research conducted in South Asia [12], South East Asia [34], and in another study of 35 LMICs [26], which showed increased risk of wasting in children with low maternal BMI. Low maternal BMI has been recognised as a key determinant of low birth weight that can lead to wasting amongst children [26,34]. Our research shows that children aged 24–59 months of mothers over 25 years had higher odds of being wasted. This finding supports a cross-sectional study conducted in Ethiopia that examined factors associated with stunting, wasting, and underweight, indicating that the highest percentage of under-five wasting was among children belonging to older women, where approximately 24% of under-five children with mothers aged 35 and older were wasted [39]. Similarly, additional analysis from our data revealed that 18% and 20.4% of children aged 24–59 months whose mothers were aged 25–34 years and 35–49 years, respectively, were wasted. This finding was in contradiction with a cross-sectional study conducted in Bangladesh and Ghana [40,41] that found that children born to younger mothers reported increased odds of being wasted. This could be explained by younger mothers being at increased risk of intrauterine growth restriction, low birth weight, and preterm birth, while older mothers have healthier children than younger mothers, because older mothers are experienced in childcare, including maintaining adequate nutrition outcomes for their children.

Children aged 0–23 months had higher odds of being wasted if they were born second or within an interval less than or equal to two years. Research suggests narrowly spaced pregnancies are among the causes triggering undernutrition amongst pregnant women in LMICs, increasing the risk of poor birth outcomes [25,42]. A study of 17 developing countries using the DHS data further provides evidence that birth spacing of at least 36 months provides protection against child undernutrition [43].

Our results showed that children aged 24–59 months and 0–59 months of mothers who made no antenatal care (ANC) visits during pregnancy were less likely to be wasted. In further sub-analysis, we found that this was driven by mothers’ age. Older women (aged 35–49 years) were more likely to use ANC services as compared to younger mothers (aged under 35 years). This finding is supported by a multistage cross-sectional study conducted in rural Lucknow, India, which reported that mothers aged 25 years and over were 19% more likely to use ANC during pregnancy than mothers younger than 25 years [44]. Similarly, a cross sectional study conducted in Indonesia that compared the use of ANC during pregnancy by adolescent girls and young women found increased maternal age was associated with more utilization of ANC services during pregnancy [45]. For example, a hospital based cross-sectional study conducted in Shanghai found that older aged women (25 years and older) were more likely to utilize ANC than younger women [46]. While research shows that access and utilization of health services, including higher coverage of ANC, has positive effects on child’s nutrition [47,48,49]. there is mixed evidence of associations between maternal age and utilization of ANC services.

Children aged 0–23 months and 0–59 months born through vaginal birth at a health facility were at higher odds of being wasted when compared to those born at home. Further sub-analysis revealed that younger women aged below 35 years delivered at home as compared to older women aged 35–49 years. Thus, mother’s age was a risk factor for child wasting, irrespective of use of available health services. Another possible explanation could be due to the enhanced role of traditional birth attendants (TBAs) during childbirth and the post-natal period of children. Especially in rural South Asia, TBAs play an important role in supporting the feeding practices, such as breastfeeding and complementary feeding [50,51].

Children aged 0–59 months who had adequate dietary diversity were at lower odds of being wasted as compared to those children with inadequate dietary diversity. This finding is consistent with a study conducted in 35 LMICs, which found poor dietary diversity associated with higher odds of wasting in children under five years of age [26]. Another study in Tanzania with children aged 6–23 months found consumption of a diverse diet was significantly associated with a reduction of wasting, and the likelihood of being wasted was found to decrease as the number of food groups consumed by children increased [52].

This study had several strengths and limitations. One of the strengths of this study was to examine factors associated with wasting among all children aged <5 years across three age groups: 0–59 months, 0–23 months, and 24–59 months. The study was population-based with a high response rate of an average of 97% for children and utilized the most recent nationally representative data in the five countries in South Asia. Despite these strengths, this study has limitations worth highlighting. First, as cross-sectional data were used to identify the factors of wasting, the study was unable to establish a causal relationship between the study characteristics and child wasting. Second, recall and measurement errors may have overestimated or underestimated the findings of this study, as data regarding some of the study factors were obtained from mothers up to five years after childbirth. Thirdly, maternal and child factors used in this study were limited to those collected by DHS. Maternal factors, such as maternal diabetes, which could be a contributing factor, were not part of the analysis. Pediatric diseases were only limited to childhood illness of diarrhea and fever and did not include other diseases, such as asthma or chicken pox. Finally, our study does not include measurement of bilateral edema and birth weight because data on birth weight are not available in the 2014 BDHS.

## 5. Conclusions

South Asia continues to host the largest population of wasted children in the world with wasting prevalence above 15%, making it a ‘critical public health’ emergency [5]. Our research highlights the most significant factors associated with child wasting for children <5 years; it contributes to the growing body of literature on child wasting while also contradicting some previous findings. Our research findings strongly indicate the need of more research on child wasting in South Asia. This is specifically crucial in order to achieve the global target to reduce wasting prevalence to <5% by 2025. A significantly improved policy environment is required that will enable scale-up of preventive and curative interventions with high impact on wasting incidence and nutrition outcomes, with a focus on children aged 0-23 months that have the highest wasting prevalence.

Our findings suggest that deliberate efforts must be made to address wasting in the region despite the rapid economic growth; improved access to water, sanitation and hygiene; food security; and poverty reduction. The variations of wasting prevalence across the South Asian countries suggest the need for targeted interventions to address country specific predictors, along with region specific interventions to address the common factors. To achieve rapid decline in wasting prevalence, targeted interventions are required for women of reproductive age, including adolescent girls, as well as all children aged <5 years with a specific focus on children under 2 years of age. There is an urgent need to implement nutrition-specific and nutrition sensitive interventions [18]. Nutrition specific interventions will address the immediate determinant’s nutrition, including adequate food and nutrient intake; feeding, caregiving, and parenting practices; and low burden of infectious diseases. Nutrition-sensitive interventions will address the underlying determinant’s nutrition and development including food security; adequate caregiving resources at the maternal, household, and community levels; and access to health services and a safe and hygienic environment—and incorporate specific nutrition goals and actions. Nutrition-sensitive interventions can serve as delivery platforms for nutrition-specific interventions and help enhance the effectiveness, scale, and coverage [18].

## Figures and Tables

**Figure 1 ijerph-18-04578-f001:**
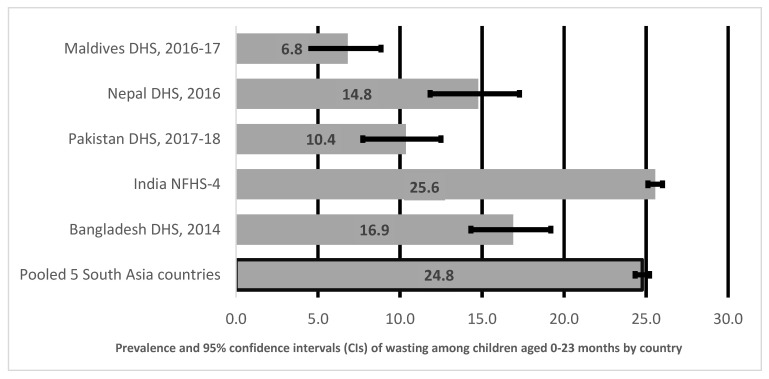
Prevalence and 95% confidence intervals (CIs) of wasting in children aged 0–23 months in South Asia.

**Figure 2 ijerph-18-04578-f002:**
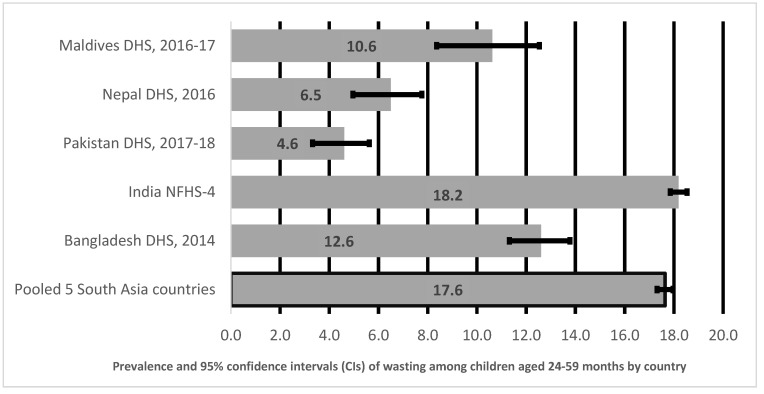
Prevalence and 95% confidence intervals of wasting in children aged 24–59 months in South Asia.

**Figure 3 ijerph-18-04578-f003:**
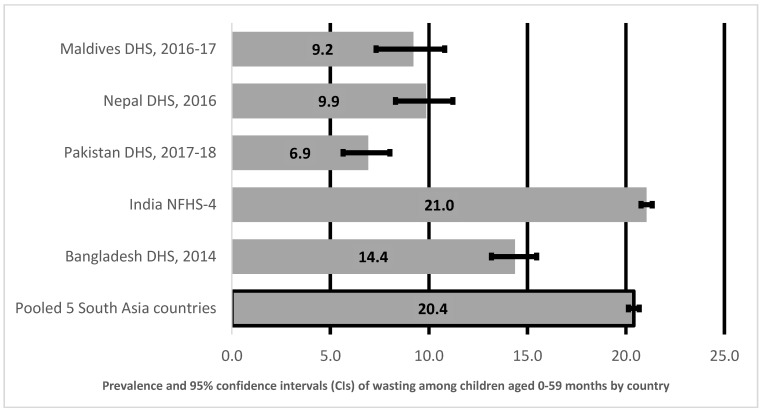
Prevalence and 95% confidence intervals of wasting in children aged 0–59 months in South Asia.

**Table 1 ijerph-18-04578-t001:** Characteristics of parents and children aged 0–59 months in five South Asia countries 2014–18.

	N	%	N *	% *
**IMMEDIATE FACTORS**				
**Dietary diversity score**				
<4 food inadequate	502,720	89.1	261,538	90.8
4+ food adequate	61,799	11.0	26,662	9.3
**Initiation of breastfeeding ^^**				
More than 1 h	98,960	55.6	54,004	57.8
Within 1 h	78,999	44.4	39,360	42.2
**Currently breastfeeding ^^**				
Yes	152,577	85.7	82,092	87.9
No	25,382	14.3	11,272	12.1
**Duration of breastfeeding ^^**				
Up to 12 months	104,612	60.3	54,918	60.6
>12 months	68,841	39.7	35,758	39.4
**Had diahrrea recently**				
No	497,294	91.6	248,960	90.7
Yes	45,449	8.4	25,432	9.3
**Had fever in last two weeks**				
No	489,556	87.0	246,288	85.8
Yes	73,202	13.0	40,904	14.2
**Vitamin A supplement**				
Yes	330,858	61.5	146,375	54.0
No	207,088	38.5	124,667	46.0
**Vaccination**				
No				
Yes **				
**Child’s age in months**				
0 to 5	46,475	8.2	26,459	9.2
6 to 11	57,785	10.2	29,444	10.2
12 to 17	57,219	10.1	29,097	10.1
18 to 23	56,863	10.1	28,191	9.8
24 to 29	59,833	10.6	28,712	10.0
30 to 35	54,883	9.7	28,490	9.9
36 to 41	61,009	10.8	30,288	10.5
42 to 47	56,638	10.0	29,601	10.3
48 to 53	58,309	10.3	28,799	10.0
54 to 59	55,503	9.8	29,119	10.1
**Sex of child**				
Male	298,109	52.8	149,820	52.0
Female	266,409	47.2	138,380	48.0
**UNDERLYING FACTORS**				
**Mother’s characteristics**				
**Mother’s age**				
15–24	203,772	36.1	94,191	32.7
25–34	315,204	55.8	163,376	56.7
35–49	45,543	8.1	30,633	10.6
**Maternal age at child’s birth**				
less than 20	85,396	15.1	36,792	12.8
20–29	404,288	71.6	202,997	70.4
30–39	71,155	12.6	44,975	15.6
40+	3679	0.7	3436	1.2
**Maternal height**				
≥160 cm	39,546	7.4	21,122	7.7
155–159	104,417	19.6	54,914	20.0
150–154	179,901	33.8	93,376	34.0
145–149	141,057	26.5	73,505	26.8
<145 cm	68,079	12.8	31,462	11.5
**Maternal BMI (kg/m^2^)**				
25+	89,949	16.9	48,632	17.8
18.5–25	341,610	64.2	184,644	67.4
≤18.5	100,822	18.9	40,768	14.9
**Birth order**				
1	234,944	41.6	105,439	36.6
2–4	298,296	52.8	157,198	54.5
≥5	31,278	5.5	25,563	8.9
**Birth interval (preceding)**				
no previous birth	236,359	41.9	106,197	36.9
<24 months	87,271	15.5	47,999	16.7
>24 months	240,798	42.7	133,924	46.5
**Combined birth rank and birth interval**				
1st birth rank	234,944	41.6	105,439	36.6
2nd/3rd birth rank, more than 2 years	209,159	37.1	110,447	38.3
2nd/3rd birth rank, less than or equal to 2	89,137	15.8	46,751	16.2
4th birth rank, more than 2 yrs interval	21,728	3.8	18,033	6.3
4th birth rank, less than or equal to 2	9550	1.7	7530	2.6
**Mother’s marital status**				
Currently married	557,795	98.9	283,875	98.6
Formerly married ^$^	6246	1.1	3934	1.4
**Working status**				
Not working	539,465	95.6	273,808	95.0
Working	25,053	4.4	14,389	5.0
**Mother’s education**				
No education	132,099	23.4	90,502	31.4
Primary	81,114	14.4	43,428	15.1
Secondary or higher	351,305	62.2	154,270	53.5
**Paternal occupation**				
Non-agriculture	96,051	17.0	50,218	17.4
Agriculture	30,003	5.3	19,651	6.8
Not working	438,244	77.7	218,182	75.7
**Power over earnings (Woman has money autonomy)**				
By husband alone	494,969	87.7	248,523	86.2
Woman alone or joint decision	69,549	12.3	39,677	13.8
**Power over household decision making**				
By husband alone	464,605	82.3	233,624	81.1
Woman alone or joint decision	99,913	17.7	54,576	18.9
**Woman has health care autonomy**				
By husband alone	477,134	84.5	240,663	83.5
Woman alone or joint decision	87,384	15.5	47,537	16.5
***Household***				
**Pooled household wealth index**				
Poorest	78,060	13.8	57,642	20.0
Poorer	83,593	14.8	57,641	20.0
Middle	99,919	17.7	57,637	20.0
Richer	140,627	24.9	57,641	20.0
Richest	162,320	28.8	57,639	20.0
**Source of drinking water**				
Not improved	58,376	10.3	42,551	14.8
Improved	506,142	89.7	245,649	85.2
**Type of toilet facility**				
Improved	319,836	56.7	142,679	49.5
Unimproved	244,603	43.3	145,453	50.5
***Access to services***				
***Healthcare utilisation factors***				
**Place of delivery**				
Home	101,719	18.2	73,821	25.9
Health Facility	457,175	81.8	211,213	74.1
**Mode of delivery**				
Non-caesarean	421,582	75.5	244,091	85.7
Caesarean section	136,657	24.5	40,674	14.3
**Combined mode and place of delivery**				
Home	101,179	18.1	73,645	25.9
Vaginal birth	320,398	57.4	170,442	59.9
Caesarean and Health Facility	136,656	24.5	40,672	14.3
**Delivery Assistance**				
Health professional	404,219	72.7	185,946	65.8
Traditional birth attendant	48,560	8.7	34,565	12.2
Other untrained ^&^	103,133	18.6	62,303	22.0
**Antenatal clinic visits**				
≥ 8	87,381	16.3	24,789	8.9
4 to 7	139,816	26.1	66,440	23.8
1 to 3	110,784	20.6	71,963	25.7
None	198,638	37.0	116,360	41.6
***Media***				
**Reads newspaper**				
Not all	357,336	63.3	201,210	69.8
Yes ^++^	207,160	36.7	86,966	30.2
**Listening to radio**				
Not all	481,048	85.2	244,719	84.9
Yes ^++^	83,471	14.8	43,481	15.1
**Watches television**				
Not all	136,315	24.2	95,183	33.0
Yes ^++^	428,203	75.9	193,017	67.0
**BASIC FACTORS**				
**Countries**				
Bangladesh	14,853	2.6	7886	2.7
India	519,243	92.0	259,627	90.1
Maldives	4708	0.8	3085	1.1
Nepal	6739	1.2	4994	1.7
Pakistan	18,975	3.4	12,608	4.4
**Type of place of residence**				
Urban	251,655	44.6	72,552	25.2
Rural	312,863	55.4	215,648	74.8

^^ analysis were restricted to 0–23 months. N* = weighted count, % * = weighted percent; % = unweighted percent. $ = formerly in union/living with a man, never in union [includes married gauna]; ^&^ assistance from friends, relatives, neighbours, no one and others; ++ = less than once a week and at least once a week. ** Yes if the child received a Bacillus Calmette–Guerin vaccination against tuberculosis; 3 doses of diphtheria, pertussis, and tetanus vaccine; ≥3 doses of polio vaccine; and 1 dose of measles vaccine and No otherwise. The color marking is to highlight the labels (characteristics) of the values in the table.

**Table 2 ijerph-18-04578-t002:** Factors associated with wasting in children aged 0–23 months in five South Asian countries (2014–2018).

Variables	0–23 Months							
	OR	95%CI		*p*-Value	AOR	95%CI		*p*-Value
**Countries**								
Maldives	1.00				1.00			
India	4.20	2.22	7.97	<0.001	3.63	1.92	6.87	<0.001
Bangladesh	2.15	0.83	5.54	0.113	1.97	0.76	5.12	0.162
Nepal	2.00	0.99	4.04	0.053	1.70	0.85	3.39	0.131
Pakistan	1.20	0.58	2.48	0.632	1.03	0.50	2.13	0.928
**Maternal height**								
≥160 cm	1.00				1.00			
155–159	1.31	0.93	1.86	0.124	1.29	0.92	1.80	0.141
150–154	1.36	1.01	1.83	0.040	1.35	0.99	1.86	0.061
145–149	1.40	1.06	1.85	0.020	1.38	1.04	1.84	0.025
<145 cm	1.32	0.89	1.95	0.161	1.33	0.93	1.91	0.118
**Maternal BMI (kg/m^2^)**								
25+	1.00				1.00			
18.5–25	1.35	1.11	1.64	0.003	1.19	0.96	1.48	0.120
≤18.5	2.25	1.76	2.88	<0.001	2.02	1.52	2.68	<0.001
**Combined birth rank and birth interval**								
1st birth rank	1.00				1.00			
2nd/3rd birth rank, more than 2 years	0.84	0.66	1.07	0.152	0.82	0.67	1.02	0.074
2nd/3rd birth rank, less than or equal 2	1.07	0.84	1.37	0.565	1.00	0.81	1.26	0.933
4th birth rank, more than 2 yrs interval	1.38	1.06	1.79	0.015	1.31	1.04	1.65	0.021
4th birth rank, less than or equal to 2	0.99	0.75	1.31	0.936	0.99	0.77	1.27	0.948
**Combined mode and place of delivery**								
Caesarean and Health Facility	1.00				1.00			
Vaginal and Health Facility	1.50	1.17	1.94	0.002	1.36	1.04	1.76	0.023
Home	1.36	1.07	1.72	0.011	1.29	1.01	1.64	0.042
**Child age in months**								
0 to 5 months	1.00				1.00			
6 to 11 months	0.67	0.52	0.86	0.002	0.64	0.51	0.82	<0.001
12 to 17 months	0.54	0.40	0.74	<0.001	0.51	0.39	0.69	<0.001
18–23 months	0.47	0.34	0.65	<0.001	0.45	0.33	0.61	<0.001
**Pooled Household wealth index**								
Poorest	1.00				1.00			
Poorer	0.73	0.64	0.83	<0.001	0.78	0.68	0.89	<0.001
Middle	0.72	0.62	0.83	<0.001	0.80	0.69	0.93	0.003
Richer	0.76	0.60	0.96	0.022	0.91	0.73	1.13	0.397
Richest	0.72	0.52	0.98	0.037	0.98	0.70	1.38	0.909

OR = unadjusted odd ratios (OR), AOR = adjusted OR.

**Table 3 ijerph-18-04578-t003:** Factors associated with wasting in children aged 24–59 months in five South Asian countries (2014–2018).

Variables	24–59 Months							
	OR	95%CI		*p*-Value	AOR	95%CI		*p*-Value
**Countries**								
Maldives	1.00				1.00			
India	1.99	1.29	3.08	0.002	1.83	1.06	3.16	0.029
Bangladesh	1.19	0.75	1.88	0.461	1.15	0.66	2.02	0.615
Nepal	0.59	0.36	0.97	0.037	0.53	0.29	0.96	0.036
Pakistan	0.33	0.19	0.58	<0.001	0.36	0.20	0.67	0.001
**Mother’s age**								
15–24	1.00				1.00			
25–34	1.14	0.95	1.36	0.170	1.24	1.02	1.51	0.034
35–49	1.32	0.91	1.92	0.145	1.49	1.01	2.20	0.044
**Maternal BMI (kg/m^2^)**								
25+	1.00				1.00			
18.5–25	1.68	1.15	2.45	0.007	1.82	1.29	2.55	0.001
≤18.5	2.39	1.60	3.55	<0.001	2.54	1.83	3.54	<0.001
**Antenatal clinic visits**								
≥8	1.00				1.00			
4 to 7	0.85	0.59	1.22	0.376	0.83	0.59	1.18	0.305
1 to 3	0.78	0.53	1.14	0.194	0.72	0.50	1.04	0.077
None	0.73	0.52	1.03	0.073	0.68	0.49	0.94	0.021
**Watches television**								
Not all	1.00				1.00			
Yes	1.05	0.90	1.22	0.561	1.25	1.06	1.46	0.006
**Pooled household wealth index**								
Poorest	1.00				1.00			
Poorer	1.03	0.90	1.17	0.692	1.03	0.90	1.18	0.629
Middle	0.98	0.83	1.15	0.791	0.93	0.77	1.13	0.472
Richer	0.76	0.64	0.89	0.001	0.72	0.61	0.86	<0.001
Richest	0.88	0.68	1.15	0.366	0.89	0.71	1.12	0.328

OR = unadjusted odd ratios (OR), AOR = adjusted OR.

**Table 4 ijerph-18-04578-t004:** Factors associated with wasting in children aged 0–59 months in five South Asian countries (2014–2018).

Variables	0–59 Months							
	OR	95%CI		*p*-Value	AOR	95%CI		*p*-Value
**Countries**								
Maldives	1.00				1.00			
India	2.66	1.77	4.01	<0.001	2.10	1.34	3.27	0.001
Bangladesh	1.51	0.92	2.48	0.101	1.42	0.85	2.37	0.177
Nepal	1.04	0.66	1.65	0.851	0.82	0.50	1.34	0.438
Pakistan	0.61	0.38	0.99	0.043	0.57	0.35	0.93	0.024
**Maternal height**								
≥160 cm	1.00				1.00			
155–159	1.32	1.03	1.70	0.030	1.32	1.01	1.71	0.039
150–154	1.28	1.03	1.58	0.024	1.24	0.99	1.56	0.062
145–149	1.31	1.04	1.64	0.019	1.25	0.98	1.59	0.071
<145 cm	1.11	0.79	1.57	0.538	1.14	0.85	1.51	0.386
**Maternal BMI (kg/m^2^)**								
25+	1.00				1.00			
18.5–25	1.59	1.25	2.03	<0.001	1.47	1.18	1.83	0.001
≤18.5	2.47	1.88	3.23	<0.001	2.18	1.72	2.77	<0.001
**Antenatal clinic visits**								
≥8	1.00				1.00			
4 to 7	0.88	0.70	1.09	0.242	0.85	0.68	1.07	0.174
1 to 3	0.93	0.72	1.20	0.588	0.83	0.64	1.08	0.167
None	0.73	0.59	0.91	0.005	0.76	0.60	0.95	0.015
**Combined mode and place of delivery**								
Caesarean and Health Facility	1.00				1.00			
Vaginal and Health Facility	1.41	1.14	1.74	0.002	1.29	1.07	1.55	0.008
Home	1.25	0.99	1.58	0.059	1.15	0.96	1.39	0.126
**Dietary diversity score**								
<4 food inadequate	1.00				1.00			
4+ food adequate	0.86	0.68	1.09	0.218	0.73	0.58	0.91	0.004
**Child age in months**								
0–23 months	1.00				1.00			
24–59 months	0.66	0.58	0.75	<0.001	0.68	0.59	0.78	<0.001
**Pooled Household wealth index**								
Poorest	1.00				1.00			
Poorer	0.89	0.82	0.96	0.004	0.90	0.83	0.98	0.019
Middle	0.86	0.76	0.97	0.011	0.89	0.78	1.03	0.112
Richer	0.76	0.66	0.88	<0.001	0.82	0.70	0.95	0.010
Richest	0.80	0.65	0.98	0.034	0.97	0.79	1.20	0.792

OR = unadjusted odd ratios (OR), AOR = adjusted OR.

## Data Availability

Data for this study were obtained from a password-enabled Measure Demographic Health Survey (DHS) website.

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
