# Peer review of "Wasting and Associated Factors among Children under 5 Years in Five South Asian Countries (2014–2018): Analysis of Demographic Health Surveys"

_ijerph, 2021, doi:10.3390/ijerph18094578_

Round 1
Reviewer 1 Report
This paper aimed to identify factors associated with wasting among children aged 0–23 months, 24–59 months, and 0–59 months in South Asia. A weighted sample of 564,518 children aged 0–59 months from the most recent Demographic and Health Surveys (2014–2018) was collected from five countries in South Asia. Multiple logistic regression analyses that adjusted for clustering and sampling weights were used to examine associated factors. The authors found that wasting prevalence was higher for children aged 0-23 months (25%) as compared to 24 -59 months (18%), with variations in prevalence across the South Asian countries. The most common factor associated with child wasting was maternal BMI [adjusted odds ratio (AOR) for 0-59 months = 2.18; 95% CI: (1.72, 2.77)]. Other factors included maternal height and age, household wealth index, birth interval and order, children born at home and access to antenatal visits. According to the study findings, they concluded the need for nutrition specific and sensitive interventions focused on women, including adolescents and children under 2 years of age. In general, the manuscript is well-written. However, there are some minor points that could be clarified for a better understanding of the readers.
- In my opinion, children wasting during 0-23 months is more related to neonatal physical condition and maternal condition including maternal health and nutritional status. This is due to breast feeding among many of the mothers and fetuses aged 0-23 months. In contrast, children wasting above 23 months seems more related to individual nutritional and health status themselves, as well as the family household wealth index. Therefore it may be beneficial for the study to consider the factors birth body weight of the fetuses and maternal health (such as maternal diabetes) in the analysis of children wasting during 0-23 months, and to lay more emphasis on the factors including pediatric diseases and family household wealth index, if possible. If doing this is difficult or data are unavailable, the authors may consider to put these concerns as the limitation of the studies.
- The style of the references are inconsistent, please make changes for the references to conform to the requirement of the journal.
- minor point: The N(%) in Table 1 -> (%) can be removed to avoid redundancy.
Reviewer 2 Report
This is an important paper as there is a need to accelerate the fight against wasting in South Asia region. The paper presents the picture of current situation and demonstrates the severity of the situation. The paper is well written but the discussion and conclusion can be improved by focusing on what the introduction, the methods and results sections have highlighted. I have only few specific comments that can be found below:
- The phrase below needs to be completed. The wasting definition is not complete.
“Wasting, weight-for-height, in children poses a serious threat to child survival and 35 development, with the heightened risk of mortality (1)”.
- Please use the word “independent” rather than “significant” when referring to results of multivariate logistic regression
- Reference group for BMI. I suggest to use the 19-<25 as the reference category. It is very strange to use a category considered as pathologic as the reference group. Also, it has been suggested that maternal obesity increases the risk of having a wasted in the baby.
- Similar comment as above for the combined mode of birth and place of delivery. Vaginal delivery at the health facility should be the reference category.
- Why using different categorization of maternal BMI in table 2 and table 3. It should be the same. The categories must include all the values and be mutually exclusive. (25+, 18.5 -<25.0 and <18.5)
- Some announced discussion points are not covered such us the point on complementing/improving available information or the improvement in available information by analyzing the 0 to 24 months and the 24 to 59 months separately.
- The recommendation of the age group to target with interventions found in the abstract should mirror that in the conclusion. Both age groups have very high prevalence of wasting.
Reviewer 3 Report
Thank you for submitting your important article describing the prevalence of wasting and associated factors at immediate, underlying and basic levels, among children under 5 years in South Asian countries.
Minor comments:
A relationship was found between a relatively low height of the mother and wasting in children. However, the relationship was not observed/significant for the mothers with the lowest height (defined as short maternal height : <145 cm). It is advisable to address this/ try to explain this in the discussion.
Lines 35, 129: "wasting, weight-for-height, …" – please add "low" before weight , as wasting means – low weight- for- height , so that the definition will be clearer to the readers.
Please correct the following proofreading errors:
Line 196: "factors associated with wasting at (p<0.05)". It is recommended to remove the brackets.
Line 196: "factors in in the…" one "in" should be removed.
Table 1: The (%) should be removed from the column header N(%), because it is indicated in a separate column.
Line 252: "Mother having a mother…" please correct.
Line 256: an increase
Please remove the dot at the end of line 73, and add dot in line 257 after "years".
Line 334: "that show" (written by mistake twice)
Line 441: it should be 0-23 months (not years)
